# COVID-19 and Fungal Diseases

**DOI:** 10.3390/antibiotics11060803

**Published:** 2022-06-15

**Authors:** Kyoung-Ho Oh, Seung-Hoon Lee

**Affiliations:** Department of Otorhinolaryngology-Head and Neck Surgery, Korea University Ansan Hospital, Ansan 15355, Korea; ohkyoungho@korea.ac.kr

**Keywords:** antifungal therapy, COVID-19, fungal infection, fungi, immunosuppressants

## Abstract

Coronavirus Disease-2019 (COVID-19) can cause secondary bacterial and fungal infections by affecting the expression of pro-inflammatory markers, such as tumor necrosis alpha and certain cytokines, as well as the numbers of CD_4_ and CD_8_ cells. In particular, in the head and neck, various fungal species are naturally present, making it the main route of secondary infection. It is difficult to clearly distinguish whether secondary infection is caused by COVID-19 directly or indirectly as a result of the immunocompromised state induced by drugs used to treat the disease. However, the risk of fungal infection is high in patients with severe COVID-19, and lymphopenia is observed in most patients with the disease. Patients with COVID-19 who are immunosuppressed or have other pre-existing comorbidities are at a significantly higher risk of acquiring invasive fungal infections. In order to reduce morbidity and mortality in these patients, early diagnosis is required, and treatment with systemic antifungal drugs or surgical necrotic tissue resection is essential. Therefore, this review aimed to examine the risk of fungal infection in the head and neck of patients with COVID-19 and provide information that could reduce the risk of mortality.

## 1. Introduction

Coronavirus Disease-2019 (COVID-19), which is caused by the novel severe acute respiratory syndrome coronavirus 2 (SARS-CoV-2), first appeared on 6 November 2019 and was designated as a pandemic on 11 March 2020 by the World Health Organization (WHO). As of 24 April 2022, the cumulative number of confirmed cases worldwide was 505,817,953, and the number of coronavirus-related deaths was 6,213,876 [1]. SARS-CoV-2 belongs to the family Coronaviridae, which are enveloped, single-stranded ribonucleic acid (RNA) viruses. Coronaviruses are classified into the following four groups: α-, β-, γ-, and δ-coronavirus. Of these, SARS-CoV-2 belongs to the β-coronavirus group and is similar to bat CoV and SARS-CoV-1 [2]. SARS-CoV-2 expresses a specific spike glycoprotein that has a strong binding affinity to the angiotensinogen-converting enzyme 2 (ACE2) receptors; this binding affinity is approximately 20 times higher than that of SARS-CoV-1 [3]. SARS-CoV-2 infection results in a highly contagious illness that is transmitted through the respiratory droplets of infected patients, including asymptomatic carriers, either directly or indirectly [4]. Although most patients with COVID-19 are treated with conservative management, approximately 14% experience dyspnea and 2.3% die [5]. SARS-CoV-2 infection can cause serious damage to the upper and lower respiratory tracts and can lead to secondary infection from bacteria, other viruses, and fungi through immune system modulation [6]. One study reported a confirmed decrease in lymphocytes in 85% of patients with COVID-19 [7]. In particular, patients with severe COVID-19 symptoms have a higher incidence of co-infection due to a decrease in both T helper 2 (Th2) and T helper 1 (Th1) cell responses [8]. Patients with underlying diseases such as uncontrolled diabetes mellitus (DM) and structural lung disease, those undergoing long-term treatment with antibiotics, and patients with immune dysregulation caused by corticosteroids are more likely to develop invasive fungal infections [9,10]. Similarly, patients with severe COVID-19 who are on mechanical ventilation in the intensive care unit (ICU) are more susceptible to bacterial or fungal nosocomial infections [11]. In these patients, mucormycosis and aspergillosis of the paranasal sinus, as well as oral candidiasis, are more common [12]. Sepsis associated with COVID-19 damages the mucosal barrier of the head and neck, which can lead to fatal fungal infections [13], and cases of fungal infections affecting the head and neck of patients with COVID-19 have been reported. Therefore, it is important to understand the characteristics of these secondary infections and their etiology to ensure optimal patient management. The aim of this review was to investigate and summarize reports of fungal diseases of the head and neck region that occur after COVID-19 infection to provide information that could improve patient outcomes.

## 2. COVID-19 and Fungal Infection

Several studies have reported that patients with COVID-19 are more likely to experience co-infection or superinfection. In January 2020, just before COVID-19 began spreading worldwide, Chen et al. reported a fungal co-infection rate of 2% (2 of 99 patients were infected with Candida albicans or Candida glabrata) [14], and Yang et al. reported a rate of 5.7% (3 of 52 patients were infected with *Aspergillus flavus*, *Aspergillus fumigatus*, or *Candida albicans*) [15]. Subsequently, in February 2020, Zhu et al. reported that co-infection was found in 94.2% of patients with COVID-19, 91.8% of whom had bacterial co-infections, such as *Streptococcus pneumoniae*, *Klebsiella pneumoniae*, or *Haemophilus influenzae*, and 23.3% of whom experienced fungal co-infection. There was a higher incidence of fungal co-infection in patients with COVID-19 who had more severe illness, with rates of 23.3% for Aspergillus, 2.5% for Mucor, 0.8% for Candida, and 0.4% for Cryptococcus infections [16]. These fungal co-infections were speculated to have been transmitted via catheters due to a decrease in host immunity. Since then, various studies have reported secondary infections caused by multidrug-resistant pathogens [8], and there have been reports of a high incidence of invasive pulmonary aspergillosis in particular [17]. COVID-19-associated pulmonary aspergillosis (CAPA) is a disease that requires long-term treatment in an ICU due to the high mortality rate [17]; in France, 33% of patients with COVID-19 in the ICU were affected by CAPA, and in Germany, it affected up to 26% of patients [18,19]. However, Kula et al. reported that the prevalence of invasive mold disease from CAPA was only about 2% based on a systematic review of postmortem studies involving autopsies of patients with COVID-19, and they mentioned the possibility of overdiagnosis of CAPA [20]. According to the Randomized Evaluation of COVID-19 Therapy (“RECOVERY”) study of the US National Institutes of Health, the use of corticosteroids is recommended only for patients receiving supplemental oxygen or ventilation, and is not recommended for patients with mild symptoms because of the risk of secondary infection [21]. However, inappropriate overuse of corticosteroids suppresses immunity, increases the length of stay in the ICU, and induces secondary infection [22].

### 2.1. Mucormycosis in the Head and Neck

Mucormycosis is a rare disease caused by inhalation of spores into the paranasal sinuses of susceptible individuals. Despite its rarity, upon exposure, it is highly likely to spread rapidly throughout the body through angioinvasion, leading to vascular thrombosis that can cause tissue necrosis, making it a disease with a very high fatality rate, ranging from 33.3% to 80% [21,23]. Although mucormycosis is common in patients with DM, it can also occur in those with weakened immune functions, such as in cases of neutropenia, or even in healthy individuals [23]. Fungal spores enter the tissues of immunocompromised patients with impaired phagocytic function and germinate as mycelium, which causes clinical symptoms. Once the fungal spores enter, they can germinate in the nasal passages, paranasal sinus, palate, orbit, and even in the brain, sometimes leading to death (Figure 1) [24]. Mucormycosis is usually accompanied by a dark lesion in the hard palate or nasal mucosa alongside symptoms that include fever, inflammation of the nasosinus, or edema on one side of the face [25]. Mucormycosis occurs in a variety of forms. In children, gastrointestinal mucormycosis is common, as is rhinocerebral mucormycosis in patients with DM or those who are immunosuppressed, and in patients undergoing chemotherapy, disseminated mucormycosis and pulmonary mucormycosis are frequent occurrences [25]. In particular, rhinocerebral mucormycosis, which occurs mainly in the head and neck region, can develop in the absence of other symptoms, except for facial pain, or without the characteristic black lesions in patients with COVD-19 who are administered systemic corticosteroids over a long period of time [26]. The incidence of COVID-19-associated mucormycosis (CAM) has also increased with the spread of COVID-19, especially in 2021; for example, there has been an explosive increase in the number of patients with CAM in India, with more than 2000 such patients presenting in the state of Maharashtra in the month of May [27]. This was presumably influenced by the effective transmission of small sporangiospores of Mucorales due to uncontrolled DM in some patients and the dry climate [28]. In general, the incidence of mucormycosis does not differ according to sex; however, in the case of CAM, it occurs more frequently in men (73.9%) [29]. In CAM, like in other types of mucormycosis, the rhino-orbital or rhino-orbital-cerebral forms are encountered most frequently, accounting for 44–49% of cases [30]. In the head and neck region of patients with COVID-19, mucormycosis has been reported to occur more frequently in men than in women, although this is thought to be due to the higher prevalence of COVID-19 in men [31]. The main symptoms occurring in patients with these types of infections were fever, facial pain, periorbital edema, hemifacial swelling, ptosis, and ophthalmoplegia, and when diagnosis was delayed by about a week, mortality increased from 35% to 65% [32].

In 2019, the European Confederation of Medical Mycology (ECMM) and the Mycoses Study Group Education and Research Consortium presented global guidelines for the diagnosis and treatment of mucormycosis [25]. It is recommended to surgically remove the lesion first, and systemic administration of a first-line antifungal agent is recommended. It is also recommended to use high doses of liposomal amphotericin B in combination with IV isavuconazole or PO posaconazole [25]. Kamat et al. reported that among patients with COVID-19, all those with mucormycosis received antifungal therapy, and 50–60% of them underwent surgical debridement [5]; this was similar to the frequency of surgical treatment before the outbreak of COVID-19 [33]. Extensive resection of infected and necrotic tissues is important for patients with mucormycosis to reduce fungal burden. In such cases, histopathology should be performed to confirm the presence of mucormycosis infection. Since there is little bleeding in tissue affected by vascular thrombosis, it is preferable to continue excising the tissue until normal bleeding occurs. If the maxilla, palate, nasal cartilage, and orbital wall are extensively involved, additional surgery and reconstruction of the defect may be required. In such cases, it is recommended to perform reconstruction concurrently with the resection to reduce the exposure and tissue atrophy of the critical tissue [34,35,36]. However, despite appropriate treatment, the mortality rate due to mucormycosis in patients with COVID-19 remains high; therefore, it is essential to diagnose it as soon as possible [31].

### 2.2. Candidiasis in the Head and Neck

Candida species, including *C. albicans*, *C. glabrata*, *C. tropicalis*, and *C. krusei* are normal flora that inhabit mucosal surfaces, such as human skin and the respiratory, urinary, and digestive tracts. Patients with compromised immune function tend to develop mucosal candidiasis. Oropharyngeal candidiasis (OPC), which is mainly caused by *C. albicans* colonization, may lead to increased morbidity in these patients [37]. Candida is a fungal species commonly encountered in those with secondary infections, especially in patients with severe COVID-19, as a form of invasive candidiasis (Figure 2) [38]. Candidiasis in patients with severe COVID-19 mainly affects the oral mucosa, although candida retinitis has also been reported [39]. Pseudomembranous candidiasis mainly occurred in patients approximately 7–8 days after SARS-CoV-2 infection and was encountered more frequently in elderly patients over 50 years of age. Oral and retinal candidiasis was mainly observed in patients with DM, hypothyroidism, or cardiovascular disease, as well as in corticosteroid users or patients with COVID-19 who are being treated in the ICU, and it was often accompanied by candidiasis affecting other sites [40,41]. The fatality rate of invasive candidiasis is about 19–40%, although rates as high as 70% have been reported in ICU patients [42]. In patients with COVID-19, infection with Candida species adversely affects the prognosis [43]. *C. auris* fungal infections are known to spread easily in long-term care facilities; however, the number of reports of *C. auris* infections in acute care units for patients with COVID-19 has increased during the pandemic, presumably due to insufficient routine infection control protocols. In particular, the misuse of gloves and gowns and the lack of disinfection are suspected causes [44]. According to a report from the US Centers for Disease Control and Prevention (CDC) in which they conducted a point prevalence survey among patients admitted to the ward for COVID-19 treatment, a positive result was observed in 35 (52%) of 67 selected patients. Samples from six (17%) colonized patients were subsequently cultured and tested positive for *C. auris* infections, and eight (40%) of the selected patients died within 30 days of screening, although it is unclear whether *C. auris* contributed to their deaths [45]. A recent study showed that the mortality rate in patients with COVID-19 who had concomitant *C. auris* candidiasis was 83.3%, even when the correct antifungal medication was administered, and resistance to amphotericin B was observed in all *C. auris* cases [11]. SARS-CoV-2 infection causes changes in the immune response and the inflammatory cascade; however, the exact mechanism through which the changes in the immune response lead to COVID-19-associated candidiasis remains unclear. The neutrophil-to-lymphocyte ratio observed in patients with severe COVID-19 and the increase in monocyte-derived macrophages or the decreased expression of the HLA-DR in monocytes are speculated to be the cause [46,47].

Patients with a suspected *C. auris* infection should receive echinocandins (caspofungin, micafungin, or anidulafungin), azoles (fluconazole, voriconazole, or itraconazole), and amphotericin B. To optimize the efficacy of azoles and minimize their toxicity, therapeutic drug monitoring should be implemented [6]. Oral candidiasis can be treated with 10 mg clotrimazole or oral nystatin suspension (400,000–600,000 IU/mL) as a mouthwash [48]. Since multidrug-resistant candida infections occur frequently, prompt preventive measures should be implemented if nosocomial infection is suspected [49].

### 2.3. Aspergillosis in the Head and Neck

Aspergillosis is the second most common mycosis affecting the oral cavity and paranasal sinuses. It is most frequently caused by *Aspergillus fumigatus*, although it can also result from infection with other species, such as *Aspergillus niger*, *Aspergillus flavus*, and *Aspergillus terreus* [50]. Similar to the species that cause mucormycosis, these fungi also invade vascular tissue, causing thrombosis and infarction. Invasive aspergillosis progresses slowly, but is highly aggressive and invasive [51]. Aspergillosis of the maxillary sinus can invade the palatal artery and cause palatal infarction, sometimes spreading to the systemic circulation. There have been reports of maxillofacial aspergillosis in patients with DM who were administered corticosteroids after being diagnosed with COVID-19 [30]. However, the incidence of aspergillosis in the head and neck region is relatively low compared to that of candidiasis and mucormycosis [52]. In the case of aspergillosis in the head and neck region, similar to mucormycosis, surgical management and systemic medication must be administered in combination [53]. In invasive aspergillosis, voriconazole has been reported to be more effective than deoxycholate amphotericin B, and liposomal amphotericin B may be used in some cases [53].

## 3. Conclusions

The aim of this review was to raise awareness of the importance of early detection and treatment of fungal infections, especially mucormycosis, as the mortality rates are high in patients with COVID-19. Therefore, it is absolutely necessary to accurately identify the risk factors associated with secondary infections in each patient diagnosed with COVID-19 to ensure optimal clinical outcomes.

## Figures and Tables

**Figure 1 antibiotics-11-00803-f001:**
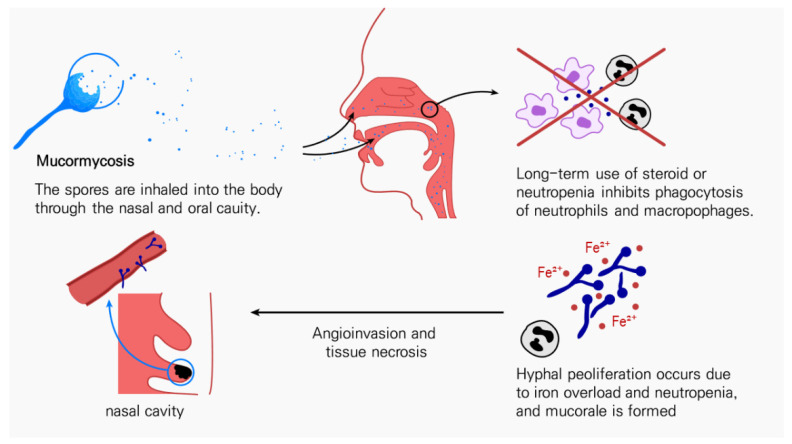
Pathogenesis of mucormycosis.

**Figure 2 antibiotics-11-00803-f002:**
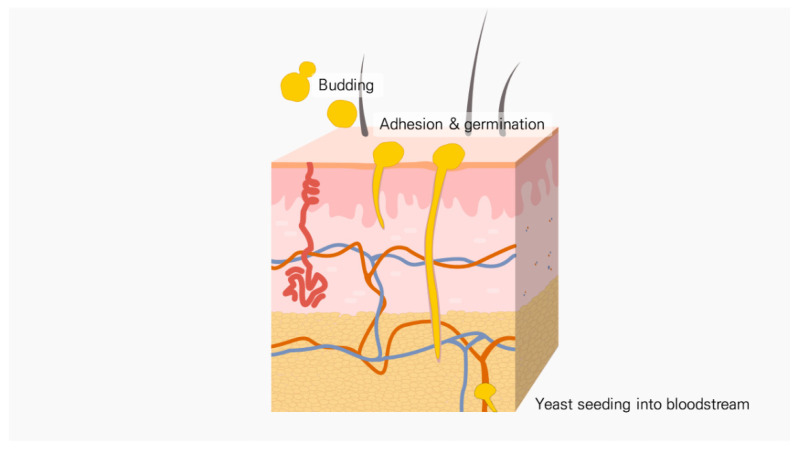
Pathogenesis of candidiasis. Candida adheres to the epithelium and then germinates. Candida degrades host protein, and the hyphae penetrate the surrounding tissue. After this, seeding in the bloodstream or colonization in the endothelium is performed.

## Data Availability

Not applicable.

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
