# Peer review of "COVID-19 and Fungal Diseases"

_antibiotics, 2022, doi:10.3390/antibiotics11060803_

Round 1

Reviewer 1 Report

The authors presented a very hot topic on COVID-19 and fungal disease, however rectification of following points is needful:

  1. The title of the review may please be changed to COVID-12 and fungal diseases.
  2. In line 10 the word severe may please be omitted in the abstract section.
  3. In line 12 the CD4 and CD8 may please be written as CD4 and CD8.
  4. The mechanism of mucormycosis and candidiasis may please be elaborated by drawing a sketch for readers easy understandability.
  5. Overall grammar needs final check by expert in English language.

Author Response

Response to Reviewers from Author

Dear Editor.

We would like to thank the reviewers for their very constructive comments. This process has assisted us in improving our manuscript. We have attempted to carefully and thoroughly address all the reviewer’s concerns.

We hope the current version of the manuscript is suitable for the publication in Antibiotics.

--------------------------------------------------------------------------------------------------------------

Reviewer 1

  • The title of the review may please be changed to COVID-12 and fungal diseases.

Answer: Thank you for your comments. The title was changed to "COVID-19 and Fungal Diseases" as the reviewer’s comment.

àCOVID-19 and Fungal Diseases in Head and Neck

  • In line 10 the word severe may please be omitted in the abstract section.

Answer: Thank you for your comments. We agree with the reviewer's opinion. As you said, "severe" has been removed.

à Severe Coronavirus Disease-2019 (COVID-19) can cause secondary bacterial and fungal infections by affecting the expression of pro-inflammatory markers, such as tumor necrosis alpha and certain cytokines, as well as the numbers of CD4 and CD8 cells.

  • In line 12 the CD4 and CD8 may please be written as CD4 and CD8

Answer: Thank you for your comments. We agree with the reviewer's opinion. We changed it as you said.

à Coronavirus Disease-2019 (COVID-19) can cause secondary bacterial and fungal infections by affecting the expression of pro-inflammatory markers, such as tumor necrosis alpha and certain cytokines, as well as the numbers of CD4 and CD8 cells.

  • The mechanism of mucormycosis and candidiasis may please be elaborated by drawing a sketch for readers easy understandability.

Answer: Thank you for your comments. As mentioned by the reviewer, we have added a picture of the pathophysiology of mucormycosis and candidiasis.

à

Figure 1. Pathogenesis of mucormycosis

Figure 2. Pathogenesis of candidiasis. Candida adheres to the epithelium and then germinates. Candida degrades host protein, and the hyphae penetrate the surrounding tissue. After this, seeding in the bloodstream or colonization in the endothelium is performed.

  • Overall grammar needs final check by expert in English language.

Answer: Thank you for your comments. We already implemented English proofreading through native speakers, but we did it again. It has been modified as follows.

à Line 52: In these patients, mucormycosis and aspergillosis of the paranasal sinus, and as well as oral candidiasis are more common

Line 149: Candida species like, including C. albicans, C. glabrata, C. tropicalis, and C. krusei are normal flora that inhabit mucosal surfaces, such as human skin and the respiratory, urinary, and digestive tracts.

Reviewer 2 Report

Review report:

This is a clinically relevant review paper that examines the risk factors and pathophysiology of invasive fungal infections in the head and neck region in patients with severe covid-19. It is further divided into three sections- mucormycosis, candidiasis and aspergillosis of head and neck region which has all been well described in patients with covid-19 disease.

Introduction:

Background to COVID-19 is laid out. Authors discuss lymphopenia secondary to COVID-19 portending a risk factor for invasive fungal infections.

Line 52: replace ‘hospital’ infections with nosocomial infections

Line 52: fungal infection pertaining to paranasal sinuses should be described

Line 56: remove the word ‘continually’

Covid-19 and fungal infection section:

Line 70: it should be Klebsiella pneumoniae, Streptococcus pneumoniae and Hemophilus influenzae

Line 82: the term should be ‘ mould’

Line 88: abuse of corticosteroids should be replaced with ‘ inappropriate overuse of steroids’

Mucormycosis in head and neck:

Line 132: omit word ‘moderate’, should be IV isavuconazole or PO Posaconazole

Line 137: replace ‘ fungal loading’ with ‘fungal burden’

Candidiasis of head and neck:

Line 152-154: this sentence should be reworded e.g invasive candidiasis has been described in patients with severe COVID-19.

There are general details regarding Candida auris listed, however, it is important to cite (if at all) any Candida auris head and neck infection in published literature

Line 182: should be ‘ HLA-DR gene’

Aspergillosis in head and neck:

Line 205: briefly discuss the common antifungal drugs used to treat invasive aspergillosis. While there are similarities in clinical presentation of head and neck aspergillosis versus mucormycosis, reasonable to comment on any differences in presentation.

Author Response

Response to Reviewers from Author

Dear Editor.

We would like to thank the reviewers for their very constructive comments. This process has assisted us in improving our manuscript. We have attempted to carefully and thoroughly address all the reviewer’s concerns.

We hope the current version of the manuscript is suitable for the publication in Antibiotics.

--------------------------------------------------------------------------------------------------------------

Reviewer 2

  • Line 52: replace ‘hospital’ infections with nosocomial infections

Answer: Thank you for your comments. As mentioned by the reviewer, we have changed it.

à Similarly, patients with severe COVID-19 who are on mechanical ventilation in the intensive care unit (ICU) are more susceptible to bacterial or fungal nosocomial infections

  • Line 52: fungal infection pertaining to paranasal sinuses should be described

Answer: Thank you for your comments. We agree with the reviewer's opinion. We changed the sentence you pointed out as follows.

à In these patients, paranasal sinus, mucormycosis of the oral cavity, and candidiasis and aspergillosis infections are more common

à In these patients, mucormycosis and aspergillosis of the paranasal sinus, and oral candidiasis are more common

  • Line 56: remove the word ‘continually’

Answer: Thank you for your comments. We agree with the reviewer's opinion. We removed the word.

à Sepsis associated with COVID-19 damages the mucosal barrier of the head and neck, which can lead to fatal fungal infections, and cases of fungal infections affecting the head and neck of patients with COVID-19 have been continuously reported.

  • Line 70: it should be Klebsiella pneumoniae, Streptococcus pneumoniae and Hemophilus influenzae

Answer: Thank you for your comments. We agree with the reviewer's opinion. We changed the sentence you pointed out as follows.

à Subsequently, in February 2020, Zhu et al. reported that co-infection was found in 94.2% of patients with COVID-19, 91.8% of whom had bacterial co-infections, such as Streptococcus pneumoniae, Klebsiella pneumoniae, or Haemophilus influenzae, and 23.3% of whom experienced fungal co-infection

  • Line 82: the term should be ‘mould’

Answer: Thank you for your comments. We agree with the reviewer's opinion. We changed the sentence you pointed out as follows.

à However, Kula et al. reported that the prevalence of invasive mould disease from CAPA was only about 2% based on a systematic review of postmortem studies involving autopsies of patients with COVID-19, and they mentioned the possibility of overdiagnosis of CAPA.

  • Line 88: abuse of corticosteroids should be replaced with ‘ inappropriate overuse of steroids’

Answer: Thank you for your comments. We agree with the reviewer's opinion. We changed the sentence you pointed out as follows.

à However, inappropriate overuse of corticosteroids suppresses immunity, increases the length of stay in the ICU, and induces secondary infection

  • Line 132: omit word ‘moderate’, should be IV isavuconazole or PO Posaconazole

Answer: Thank you for your comments. We agree with the reviewer's opinion. We changed the sentence you pointed out as follows.

à It is also recommended to use high doses of liposomal amphotericin B in combination with IV isavuconazole or PO posaconazole

  • Line 137:replace ‘fungal loading’ with ‘fungal burden’

Answer: Thank you for your comments. We agree with the reviewer's opinion. We changed the sentence you pointed out as follows.

à Extensive resection of infected and necrotic tissues is important for patients with mucormycosis to reduce fungal burden.

  • Line 152-154: this sentence should be reworded e.g invasive candidiasis has been described in patients with severe COVID-19

Answer: Thank you for your comments. We agree with the reviewer's opinion. We changed the sentence you pointed out as follows.

à Candida is a fungal species commonly encountered in those with secondary infections, especially in patients with severe COVID-19, as a form of invasive candidiasis [40]. Candidiasis in patients with severe COVID-19 mainly affects the oral mucosa, although candida retinitis has also been reported

  • There are general details regarding Candida auris listed, however, it is important to cite (if at all) any Candida auris head and neck infection in published literature

Answer: Thank you for your comments. Originally, the topic of this manuscript was COVID-19 and fungal disease in head and neck, but it has been changed to COVID-19 and fungal disease by another reviewer's opinion. Therefore, it would be alright to disclose in the article about the general matters of candida auris.

  • Line 182: should be ‘HLA-DR gene’

Answer: Thank you for your comments. We agree with the reviewer's opinion. We changed the sentence you pointed out as follows.

à The neutrophil-to-lymphocyte ratio observed in patients with severe COVID-19 and the increase in monocyte-derived macrophages or the decreased expression of the HLA-DR in monocytes are speculated to be the cause

  • Line 205: briefly discuss the common antifungal drugs used to treat invasive aspergillosis. While there are similarities in clinical presentation of head and neck aspergillosis versus mucormycosis, reasonable to comment on any differences in presentation

Answer: Thank you for your comments. We agree with the reviewer's opinion. We added the sentence.

à In invasive aspergillolsis, voriconazole has been reported to be more effective than deoxycholate amphotericin B, and liposomal amphotericin B may be used in some cases [55].

Round 2

Reviewer 1 Report

Thanks for rectifications of errors highlighted